# Approaches to Optimizing Medical Treatment Policy using Temporal Causal Model-Based Simulation

**Robert Horton**
Microsoft
rhorton@microsoft.com

**Maryam Tavakoli Hosseinabadi**
Microsoft
matavako@microsoft.com

**John-Mark Agosta**
Microsoft
john-mark.agosta@microsoft.com

## Abstract

It is notoriously difficult to draw conclusions about the effects of medical interventions from observational data, where statistical confounding is rampant. An important example is "confounding by severity" in which sicker patients receive more aggressive intervention, leading to a misleading positive correlation between stronger intervention and worsening outcome. This scenario is quite generally applicable because it represents negative feedback control, where some control mechanism responds to a change by affecting the change in the opposite direction. This leads to a causal loop: the change affects the feedback and the feedback affects the change. We employ the classic approach to breaking such loops by unrolling them in time, so that the disease severity before treatment is separated from the severity after treatment. Unrolling produces a dataset where the information about a patient is no longer contained on a single row of a dataframe, but is spread over a set of rows representing timeslices. We want to base treatment decisions on the final outcome, which is only found at the end of this set of rows. Since we are interested in outcomes that occur at a future timeslice, we borrow a term from reinforcement learning and describe our type of intervention as a "policy". Our challenge is to properly integrate temporal modeling with causal modeling on observational data so that we can deconstruct these causal loops and reach useful analytical conclusions. Here we demonstrate a suitable analytical approach with a simple toy problem, a drug dosing policy to treat the disorder arising from infection with the fictitious pathogen *Bogovirus*. We begin by writing a simple bespoke simulation program to match a given causal graph; this generates a simulated dataset where we know the ground-truth about causal interactions. Using the known correct influence graph, together with other aspects of "domain knowledge", we build causal model-based simulations of the simulated data ("simsim" models) that let us estimate the expected effects of various treatment policies on ultimate outcomes. We compare this approach to the closely-relate field of reinforcement learning, and show how they are complementary.

## 1 Introduction

This paper demonstrates how temporal causal models can be used as the basis of simulations that make it possible to optimize treatment policies in virtual experiments. We show the approach on data that has itself been generated by a simulation, so that we know the 'ground truth' causal relationships.

NeurIPS 2022 Workshop on Synthetic Data for Empowering ML Research.

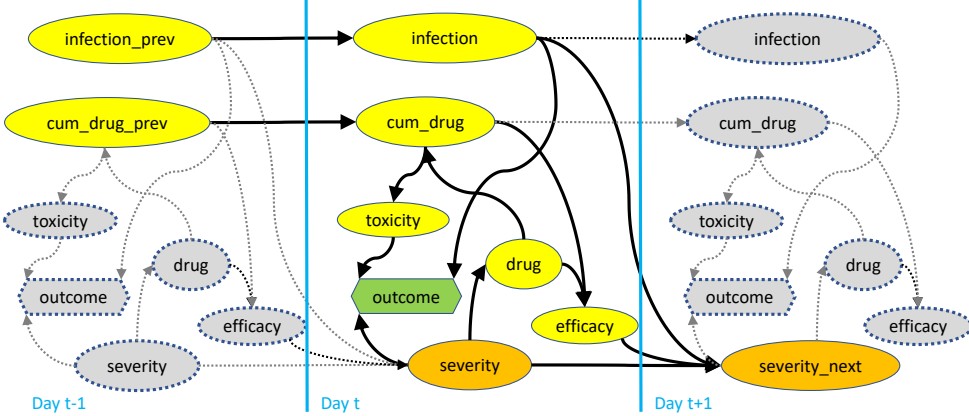

Figure 1: Unrolled causal model with sequential time steps. States explicitly included in the model are shown in color; gray ovals and dotted arrows show states and transitions that are captured in other timesteps.

Our ultimate goal is to optimize patient treatment protocols using simulations based on causal models derived from a combination of learning from patient data and incorporating domain knowledge.

## 2 BogoVirus: A Simulated Dataset

A common problem in causal modeling of medical disorders is the presence of feedback loops. A drug is given to manage a disorder, and the disorder responds to the drug; this circular feedback leads to "confounding by indication" ( Salas et al. [1999]), or its continuous-domain analog, "confounding by severity". One way to break such causal influence cycles is to unroll them in time to create a "Dynamic Bayesian Network" (Dean and Wellman [1991]); one observes the severity of the disorder in the current timestep when administering the drug, and the response to the drug is reflected in the severity of the disorder in the next timestep.

### 2.1 Primary simulation

Our model includes an underlying condition (infection with the imaginary organism *Bogovirus*); this condition will resolve over time as long as the patient does not die. Infection leads to a disorder (it could be something like a problem with blood clotting, breathing, or circulation) that must be managed because if it gets too severe it can lead to death. The disorder is managed by a drug which has a direct effect on the severity of the disorder, but becomes less effective as the level of the drug accumulates in the body. High cumulative levels of the drug can lead to fatal toxicity. The ultimate outcome is either recovery or death, but on most days neither of these things happens, and the patient just stays in the hospital for another day.

We use simulation to generate data for which we know the true causal graph. This is done by writing a simple Python program that implements the causal graph shown in Figure 1; each node is represented by a random function whose inputs are dictated by the edges leading into the node, and this set of functions are called in the order shown in the DAG to compute a row of variable values for each day of the simulated patient's illness. For each patient, this process continues until the infection process completes (leading to a "recover" outcome), or the patient succumbs to either the disorder or the drug toxicity ("die" outcome). We replicate this to simulate a population of patients.

| patient_id | cohort | day_number | infection_prev | severity | cum_drug_prev | infection | drug | cum_drug | severity_next | toxicity | outcome |
|---|---|---|---|---|---|---|---|---|---|---|---|
| 4456 | 8 | 0 | 39 | 18.000000 | 0.000000 | 42 | 0.0 | 0.000000 | 24.000000 | 0.000000 | none |
| 4456 | 8 | 1 | 42 | 24.000000 | 0.000000 | 49 | 0.0 | 0.000000 | 31.300000 | 0.000000 | none |
| 4456 | 8 | 2 | 49 | 31.300000 | 0.000000 | 57 | 0.0 | 0.000000 | 40.130000 | 0.000000 | none |
| 4456 | 8 | 3 | 57 | 40.130000 | 0.000000 | 63 | 0.8 | 0.320000 | 38.321788 | 0.001074 | none |
| 4456 | 8 | 4 | 63 | 38.321788 | 0.320000 | 71 | 0.8 | 0.512000 | 38.671956 | 0.018014 | none |
| 4456 | 8 | 5 | 71 | 38.671956 | 0.512000 | 80 | 0.8 | 0.627200 | 40.706310 | 0.060875 | none |
| 4456 | 8 | 6 | 80 | 40.706310 | 0.627200 | 87 | 0.0 | 0.376320 | 53.476941 | 0.002840 | none |
| 4456 | 8 | 7 | 87 | 53.476941 | 0.376320 | 95 | 0.8 | 0.545792 | 57.973954 | 0.026434 | none |
| 4456 | 8 | 8 | 95 | 57.973954 | 0.545792 | 103 | 0.8 | 0.647475 | 64.359519 | 0.073678 | recover |

Figure 2: Simulated data for a single patient. This is from the "RCT" dataset, where the patient has been assigned to the 0.8 dose of drug, if they are given a dose at all. Note that the drug is administered probabilistically depending on severity, so at early stages no drug is administered.

Through trial and error, we adjusted the simulation parameters so that a simple scan of a constant daily dose (see Figure 3) will show an optimum, but the peak performance is less than 100% survival, so it leaves some room for improvement.

## 2.2 Components of a medical simulation

These are the variables in the simulation:

- **infection**: records how far the patient has passed through the course of the infection (this is basically a counter for percent progress; once it passes 100 and you have not died, you recover).
- **drug**: a dose of the treatment. This is an adjustable quantity.
- **cum_drug**: The accumulated dose of the treatment drug. This is an exponential moving average, and is subject to decay over time if treatment doses are not administered.
- **efficacy**: The extent to which the drug dose affects severity; the drug becomes less effective at higher cumulative doses.
- **severity**: Quantified level of severity of the disorder caused by the infection. The worse this gets, the more likely the patient is to die.
- **outcome**: if severity gets high enough, the patient's chances of death increase. If the infection runs its course and the patient does not die, they recover.
- **toxicity**: a function of cum_drug that is much more pronounced at high cumulative dose. High toxicity leads to increased probability of death.

Variables with suffixes '_prev' or '_next' hold lagging or leading values from adjacent timeslices.

## 2.3 Simulated datasets

To simulate a randomized controlled trial (RCT) of drug dose, patients were assigned to cohorts, and each cohort received a fixed dose of the drug on each day of treatment. Figure 2 shows simulated data for a single patient episode. There is one column for each node in the model, plus housekeeping attributes `patient_id`, and `day_number` (how many days that patient has been in the hospital). The `cohort` column indicates the group into which the patient was randomized in the simulated RCT; this determines the dose of drug the patient receives.

To simulate observational data, we use the primary simulator to generate another dataset in which a random dose of drug is given to each patient each day. This randomized policy dataset is the input to the subsequent modelling tasks.

## 2.4 Finding the optimal dose

The solid red curve in Figure 3 shows survival at different doses of drug from the RCT dataset. The optimal dose (0.7 units) gives us a standard to compare with results we obtain from analytical

Figure 3: Finding the optimal dose to maximize survival. The solid red line shows the results of running a simulated Randomized Controlled Trial (RCT) in the primary simulation described in section 2; the dashed lines show the results obtained in analytical 'simsim' simulations trained on the 'observational' dataset. The viral disorder is almost uniformly fatal if untreated, drug toxicity is fatal at high cumulative doses, and there is a point where the optimal dose gives the best response. From this scan, the best outcome achieved is at a dose of 0.7 units, giving 87.1% survival.

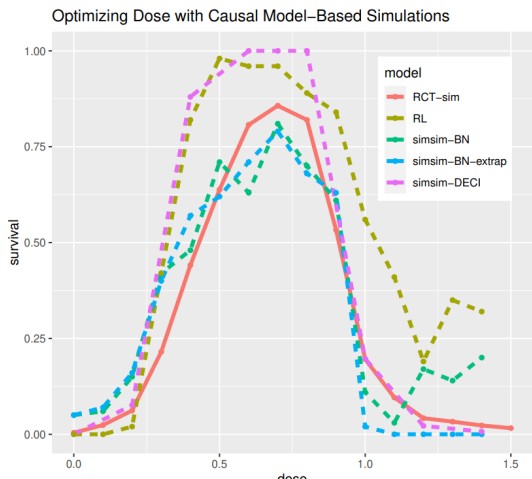

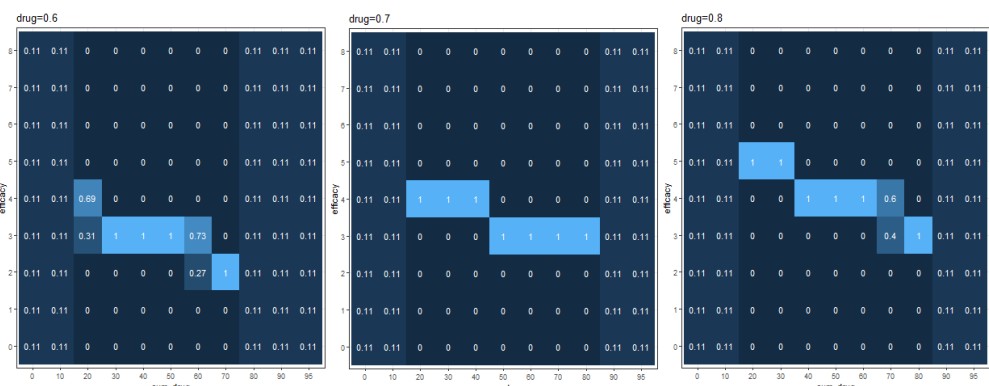

Figure 4: A portion of the Conditional Probability Table (CPT) for efficacy as a random function of dose and cumulative dose. This CPT is a three-dimensional table where each cell holds a probability, and the coordinates of the cells are the categorical values of the inputs (`drug` and `cum_drug`) and the output (`efficacy`). Note that the extreme ranges of `cum_drug` are not represented in the observational data, and the model assumes a uniform prior. These areas require editing to incorporate domain expertise for extrapolation.

simulations. Our goal in subsequent sections is to use analytical 'simsim' simulations to approximate this solid red curve without requiring an RCT.

## 3 "SimSim-BN": An Analytical Simulation Built on a Causal BayesNet Model

Rather than testing a treatment policy in the primary simulation (analogous to an experiment), we can use a causal model to build a simulation (since this is a simulation of a simulation we call this a 'simsim' model, of which we present three versions). We then use this causal simulation to perform the optimization scan. The simsim model is trained on the simulated "observational" dataset where each patient was given a random dose of the drug each day (as opposed to the dose-optimizing simulation where patients were assigned to cohorts that each received the same dose each day). This simulates the process of building a causal model from non-experimental data and using it to estimate an optimal treatment policy without actually running an experiment.

To simulate building a model from observational data, we used the known causal graph from Figure 1 to build a Bayes network whose parameters were learned from the randomized dosage dataset.

| patient_id | day_number | severity | infection | drug | cum_drug | toxicity | die | recover | prev1_severity | prev1_infection | prev1_drug | prev1_cum_drug | prev1_toxicity | prev1_die | prev1_recover |
|---|---|---|---|---|---|---|---|---|---|---|---|---|---|---|---|
| 3911 | 6 | 54.607584 | 55 | 0.8 | 0.479744 | 0.012192 | 0 | 0 | 47.696857 | 53 | 0.2 | 0.266240 | 0.000356 | 0.0 | 0.0 |
| 1326 | 10 | 46.535000 | 75 | 1.0 | 0.857256 | 0.396885 | 0 | 0 | 50.113805 | 73 | 1.4 | 0.762094 | 0.195908 | 0.0 | 0.0 |
| 451 | 0 | 23.000000 | 28 | 0.0 | 0.000000 | 0.000000 | 0 | 0 | NaN | 20 | 0.0 | 0.000000 | 0.000000 | 0.0 | 0.0 |

Figure 5: Dataset is re-framed to capture temporal dynamics on each row.

## 3.1 Extrapolating to areas not covered by the training data

Figure 4 shows the CPT for `efficacy` for three selected discrete values of `drug`. We can see that these tables have captured the relationship that the efficacy of a given dose of the drug decreases at higher levels of `cum_drug`. However, the very high or very low values of `cum_drug` (the extremes of the $x$-axis in each panel) show a uniform probability across the levels of `efficacy`. This is because these values of `cum_drug` are not represented in the random-dose dataset from which this Bayes Net was trained (that would require a consistently high or consistently low dose, which is unlikely in the random-dose dataset), and it defaults to using a uniform prior probability in these ranges. We therefore took advantage of our domain knowledge to modify the CPTs to extrapolate better to extreme values. This was done by simply filling out the uniform-probability columns at the extremes with copies of the closest non-uniform column.

We also simplified the input `infection`, which is basically a counter that keeps track of how far the patient has progressed through the course of infection, into the binary variable `infection_over` that is true if infection reaches 100 or more, and false otherwise. This greatly simplifies the probability table for `severity_next`.

## 4 "SimSim-DECI": Expert-in-the-loop Causal Inference for Intervention Optimization

Deep End-to-end Causal Inference ("DECI", Geffner et al. [2022]) [1] is an autoregressive-flow based non-linear additive noise structural equation model (SEM), which is designed to perform both causal discovery and inference, including average treatment effect (ATE) and conditional average treatment effect (CATE) estimation. DECI discovery takes prior knowledge of the graph structure as input, defined by a constraint adjacency matrix, and uses the data-driven causal calculation to estimate the graph structure and ATE between nodes.

### 4.1 Using DECI for Timeseries: A Featurization Method

*SimSim-DECI* uses the DECI model to simulate temporal records for the patients under various drug-dose policies and finds the policy for the optimum dosage level. DECI is primarily designed to digest generic types of tabular datasets. To capture the temporal dynamics of our data, we designed our framework as follows:

- *Temporal Sliding-Window featurization*: each row contains the information of the current time-step and the previous one as in $(X_{t-1}, X_t)$. Figure 5 shows what re-framed dataset looks like.

- *Temporal constraints*: Features in the current steps cannot cause features in the previous step.

- *Temporal training*: The DECI model is trained on each row as an independent sample. Since each row holds the past and current features, the connection between features of the same day is learned as well as features of two consecutive days.

- *Temporal simulation*: We simulate the information of each step by conditioning the simulation on the information from the previous step. That means for data in Figure 5, we take the values of the seven right columns as the condition for the intervention and predict the next step by sampling the trained model.

---

[1] `https://github.com/microsoft/causica/`

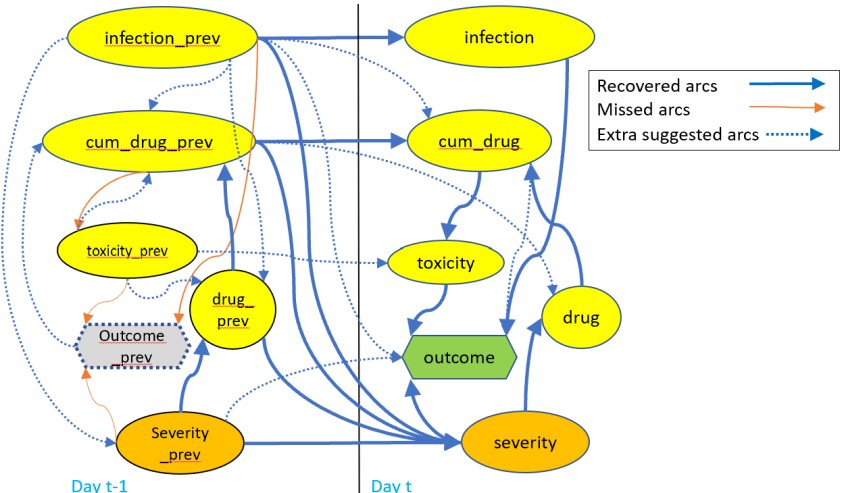

Figure 6: The learned model with causal constraints. By adding structure constraints, DECI returns a model, which misses 4 arcs, but recovers 14 arcs. It also, suggests 11 extra arcs, which are not accurate based on the original model. Note, this 2-day graph is the equivalent to the original 3-day one on Figure 1. This is possible because each feature only depends on features not more than one day away. Also, we do not need efficacy node as DECI can handle more complex inputs than the table-based Bayes net model used above.

## 4.2  Causal Discovery

DECI recovers the structure of the underlying causal graph that we can compare with the true underlying graph. Thus, we can be the expert-in-the-loop to modify the data-driven findings to improve its correspondence to the truth. We do this by re-training with additional constraints. Figure 6 shows the final graph with expert-in-the-loop constraints applied. Note that the domain expert prevented *confounding by indication* (Kyriacou and Lewis [2016]) by reversing the direction of drug-to-severity causality.

## 4.3  Simulation-based Policy Optimization

The trained model is a structural causal model (SCM) that captures the functional relations and error distributions. Hence, it is capable of estimating the expected updates following an intervention. We use this capability to synthesize the current day, having information about the previous day. We add the desired drug dose based on the candidate policies, explained in previous sections, to the interventions. To reduce estimation error, we take the average of 50 samples for the continuous variables and take the mode for the binary ones. The simulation for each patient will stop if "die" or "recover" returns True. Figure 3 demonstrates the results of different dose policies. The model captures the optimum drug dose correctly, although is less accurate at extremes where the observational data is sparse (see the "simsim-DECI" curve in Figure 3). Note that this model is too optimistic about survival rates for intermediate doses; this may be because the learned graph did not find the edge from 'cum_drug' to 'efficacy', which reduces the benefit provided by the drug at moderate doses.

## 5  Offline Reinforcement Learning: The Third Simulation Reconstruction

Without the causal knowledge conveyed by the network, optimization of the dose does poorly. We show this for purposes of comparison by taking the same randomized policy simulated observational dataset and using a comparable offline reinforcement learning (RL) method to derive a constant-dose policy comparable to the previous simulations, but ignorant of the causal structure underlying the data. Because we used a constant policy the method is degenerate, since the policy function does not depend on the current state, and might equivalently be considered just a simple Markov process.

For the RL environment, we wrapped the simulation dataset with `OpenAI Gym` (Brockman et al. [2016]), to simulate the stage transition probability distribution. Unlike the previous "simsim" reconstructions, state variables are continuous-valued, so to generate a random state value given the current state, we used a k-nearest neighbor regressor built on the `faiss` (Johnson et al. [2019]) library for similarity search. Since, even with the simulation dataset of $1M$ rows, there is no likelihood that a lookup will find a record within some epsilon of the current state, so instead we average the predicted next-stage values of the $k = 3$ nearest neighbours. Equivalently the prediction function could have been implemented as a neural-network trained on the simulation dataset. Either way, predictions are confined to the space that has been explored in the simulation data, with all the limitations this implies for an offline algorithm.

We ran a constant policy in dose increments of $0.1$ from $0$ to $1.4$, for $100$ episodes, which in almost all cases terminated in less than $20$ stages, then averaged the termination counts to estimate survival rates, as shown in Figure 3. Admittedly using a constant policy is a trivial algorithm, and cannot do better than an adaptive policy. But for purposes of comparison with the previous "sim-sims", it illustrates the consequences of the lack of causal analysis. As the figure shows, the offline-RL finds the same optimum dose, but over-estimates the survival rates at extremes. Most notably it believes high doses that accumulate toxic cumulative effects are not deleterious. Inspection of predictions as revealed in the episode traces show that, at extreme values, the offline predictor is wildly off with predicting accumulation of the drug over time. This highlights a known limitation of offline policy optimization, where even in modest dimension state-spaces the ability to explore new policy areas is severely limited by the existing data (Levine et al. [2020]). Introducing causal domain knowledge as we propose, is one remedy. It is similar in spirit to the notion of exposing a tunable user parameter to moderate the extremes of offline-generated policies, as proposed by (Swazinna et al. [2022]).

# 6   Discussion: Where do we go from here?

We plan to collaborate with medical researchers in our next iteration of this simulation and modeling process to focus on an actual disease and therapies, to reproduce the analytical logic in a more realistic setting. We have experience implementing a similar forward-inference Bayes Network in the Synthea electronic medical record simulator (Walonoski et al. [2017])[2]; porting our next version to that platform would let us develop shareable feature engineering exercises to conduct the kind of simulation-based analysis described here on data in a realistic schema.

For our next iteration we plan to use Rhino (Gong et al. [2022]), the vector auto-regressive extension of DECI, instead of bespoke temporal featurization. Rhino can learn history-dependent noise, and we look forward to exploring circumstances where this leads to better simulation performance

Causal graphs provide an opportunity to capture domain expertise; this is a modeling process that, as we have shown, blends gracefully into simulation modeling. The advantage of adding the causal aspect to the modeling process is that we have an explanation of the model's function in terms comparable to current clinical understanding, and therefore improves credibility and the extensibility of the model's results. We imagine adding a causal modelling step to extend offline simulation for an envisioned causal reinforcement learning model.

The code for this project is in Github [3].

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
