# OpenReview forum: "Approaches to Optimizing Medical Treatment Policy using Temporal Causal Model-Based Simulation"
_NeurIPS.cc/2022/Workshop/SyntheticData4ML — Neurips 2022 SyntheticData4ML_

### Official Review · Reviewer_7tQb · 2022-10-11
**Interesting paper, perhaps lacking some model details**

**Rating:** 6
**Confidence:** 3

**Review:**

I think the paper is quite interesting. Model based policy optimisation seems like a great fit to be combined with causal discovery. There were a few things that were lacking for me:
- As this is a workshop on synthetic data, and the focus on the paper is more causal modelling and policy optimisation. It would make more sense to submit elsewhere.
- How are the models learned/optimised? There seem to be little details on the actual learning process

---

### Official Review · Reviewer_RGWr · 2022-10-19
**Causal model-based simulation for policy optimization**

**Rating:** 6
**Confidence:** 4

**Review:**

This paper proposes a simsim framework where a temporal causal model is learned from simulated data and subsequently used for optimizing treatment dosing policy.

Pros:

1. Using temporal causal model for simulation is well-motivated. This work leverages causal BayesNet and DECI to serve as the causal model. The learned causal graph helps ensure that the downstream simulation is grounded.

2. Using simulation to optimize policy is also a proven approach in related fields such as reinforcement learning.

Cons:

1. A more in-depth comparison of the three approaches would be appreciated. What are the respective strengths and weakness, the assumptions and prior knowledge required?

---

### Meta-Review · Area_Chair_8qeT · 2022-10-19

**Recommendation:** Accept